# Anti-Proteolytic Peptide R7I Protects the Intestinal Barrier and Alleviates Fatty Acid Malabsorption in *Salmonella typhimurium*-Infected Mice

**DOI:** 10.3390/ijms242216409

**Published:** 2023-11-16

**Authors:** Yunzhe Su, Taotao Sun, Junhan Gao, Chenxu Zhang, Xuesheng Liu, Chongpeng Bi, Jiajun Wang, Anshan Shan

**Affiliations:** College of Animal Science and Technology, Northeast Agricultural University, Harbin 150030, China; suyunzhe98@163.com (Y.S.); suntaot@outlook.com (T.S.); 13204660356@163.com (J.G.); 13102981697@163.com (C.Z.); liuxs0909@163.com (X.L.); bnm0722@163.com (C.B.)

**Keywords:** *Salmonella typhimurium*, anti-proteolytic peptide, inflammation, intestinal barrier, fatty acid absorption

## Abstract

With a wide range of hosts, environmental adaptation, and antibiotic resistance, *Salmonella typhimurium* is one of the most common causes of food poisoning in the world. Infection with *Salmonella typhimurium* not only results in intestinal inflammation but also damages the intestinal barrier and interferes with the host’s ability to absorb nutrients. It is imperative to find alternatives to antibiotics for eradicating bacteria, reducing intestinal damage, and reestablishing nutrient absorption, especially given that antibiotics are currently prohibited. This research aims to understand the protective role of anti-proteolytic peptide R7I on the gut in the setting of *Salmonella typhimurium* infection and its impact on nutritional absorption, maybe offering an alternative to antibiotics for bacterial killing. The findings demonstrated that R7I reduced the production of inflammatory factors, including IL-6, TNF-α, and L-1β in the jejunum and decreased the expression of genes like *TLR4* and *NF-κB* in the jejunum (*p* < 0.05). R7I enhanced antioxidant capacity and preserved the antioxidant/pro-oxidant balance in the jejunum *(p* < 0.05). R7I also normalized intestinal shape and restored tight junction protein expression. Fatty acid binding protein 2 (FABP2) and fatty acid transport protein 4 (FATP4) expression in the jejunum was restored by R7I. In addition, serum-free fatty acids and lipid metabolites were significantly higher in the R7I group than in the control group (*p* < 0.05). Overall, the anti-enzyme peptide R7I maintained the healthy state of the intestine and alleviated the abnormal fatty acid absorption caused by bacterial infection.

## 1. Introduction

Salmonellosis is one of the zoonotic diseases of public health importance, posing a great threat to human health, food hygiene, and safety. The pathogen *Salmonella* is one of the most common foodborne pathogens, and globally, *Salmonella* infections cause gastroenteritis in 550 million people each year, with about 420,000 deaths [1]. Gastroenteritis is the most common manifestation of *Salmonella* infection worldwide, followed by bacteremia and enteric fever [2]. *Salmonella* infections in adults usually cause gastrointestinal illness, and children, the elderly, and other immunocompromised people are more susceptible to *Salmonella*, which can lead to death in severe cases [3,4].

*Salmonella* is a fairly common pathogen that can spread from the environment to animals and people as well as to them through food, water, infected animals (chickens, pigs, and cows), as well as their byproducts [5,6]. *Salmonella* has a complex antigenic structure and a significant number of serotypes, with 2600 now recognized [7]. The two serotypes of *Salmonella* most frequently transferred from animals to people in clinical settings are *Salmonella enteritidis* and *Salmonella typhimurium* [8]. The most dangerous of them all is *Salmonella typhimurium*. *Salmonella typhimurium* infections can range from enteritis to sepsis and systemic organ damage, and they can spread to other organs, which is one of their main characteristics [9,10]. *Salmonella typhimurium* primarily infects hosts by tainted food or drink, after which it assaults the host’s vulnerable digestive system [11]. *Salmonella typhimurium* type Ⅰ hairs enable the bacteria to cling to and colonize the small intestine’s epithelial cells during the early stages of infection. The *Salmonella typhimurium* type III secretion system subsequently enters the picture, releasing effector proteins into the intestinal epithelium such as SipA, SipB, SipC, SopE, and SigD, which exploit host signaling pathways to induce alterations in the epithelial cytoskeleton [12]. These alterations disrupt the otherwise normal epithelial brush border and induce its extension and generation of membrane folds, eventually forming a phage-like structure of vesicles, known as SCV (*Salmonella*-containing vacuole), within the host cell cytoplasm, which contributes to the survival and multiplication of *Salmonella typhimurium* in the host [13,14].

Infection with *Salmonella typhimurium* in the host triggers the immune system, leading to inflammation and the release of INF-γ, IL-1β, and TNF-α inflammatory cytokines. However, it has been discovered that incubating intestinal epithelial cells with INF-γ and TNF-α rearranges the tight junction proteins on the cells from the cell membrane to the cytoplasm, causing the intestinal epithelial barrier function to be disrupted [15]. Lipopolysaccharide (LPS) in the cell wall of *Salmonella typhimurium* can cause disruption of the intestinal epithelial barrier function and intact tight junctions during infection of the host [16,17]. Additionally, the *Salmonella typhimurium* type III secretion system raises cell permeability, which changes the way the intestinal barrier functions [18,19]. The primary phagocytes that gather in the Peyer’s nodes and mesenteric lymph nodes of the small intestine wall are neutrophils and macrophages [20]. After the host has been infected with *Salmonella typhimurium*, infiltrated neutrophils can kill extracellular bacteria, but they can also harm intestinal epithelial barrier tissue and exacerbate the inflammatory response [21]. Animals depend heavily on their intestinal barriers to protect them from pathogenic bacteria and other germs in the intestinal lumen. These barriers also ensure that the body gets the nutrients it needs to function properly. Reduced growth performance and systemic sickness may result from a damaged intestinal barrier, which may also cause loss of nutrients and the spread of dangerous microorganisms from the intestine to other organs [17,22]. The intestinal barrier of the host is harmed by *Salmonella typhimurium* infection through a variety of mechanisms, which most likely hinder nutrients from being absorbed. Not only that, but *Typhimurium* is also one of the main serotypes that cause antibiotic resistance and one of the main strains of *Salmonella* that cause multidrug resistance [23]. Therefore, the use of new antibiotic alternatives such as antimicrobial peptides, probiotics, and plant extracts to suppress bacterial infections has become a future trend [11,24]. Antimicrobial peptides (AMPs) have drawn a lot of attention as a treatment for germs that are resistant to numerous drugs. In the majority of multicellular organisms, antimicrobial peptides serve as the initial innate immune system defense [25,26]. Antimicrobial peptides serve as a crucial first line of defense for the body’s immune system. They have broad-spectrum, highly effective antibacterial activity, eliminate bacteria more quickly than conventional antibacterial drugs, and are less likely to develop drug resistance thanks to their special mechanism of binding and physical membrane disruption [24,27].

However, a critical issue with antimicrobial peptides in clinical applications is their weak resistance to protease hydrolysis [28]. The natural amino acids in the anti-proteolytic peptide R7I that our lab has developed are logically structured to avoid trypsin and chymotrypsin cleavage sites as much as feasible. Preliminary tests revealed that the anti-proteolytic peptide R7I has outstanding bactericidal and anti-enzymatic properties. In vitro, R7I demonstrated outstanding resistance to pepsin, trypsin, and pancreatic rennet. The 8-h incubation of R7I in mouse stomach and small intestine fluids did not affect the antibacterial activity of the substance [29]. R7I has also shown effective treatment of enteritis brought on by E. coli [29]. However, it was not examined in earlier research whether the anti-proteolytic peptide R7I repaired intestinal damage brought on by *Salmonella typhimurium* and whether it improved infection-induced nutrition absorption impairment. 

The emergence and spread of antimicrobial-resistant strains of *Salmonella* have become a major public safety issue over the years due to the long-term use of antibiotics. *Salmonella’s* resistance to antibiotics and the types of antibiotics tolerated are increasing [30,31]. Aside from that, fatty acids are also one of the three primary nutrients that play a significant role and are crucial for the preservation of the body’s normal vital functions, and they are mostly absorbed in the jejunum. Our team’s motivation for conducting this research was the issue of medication resistance in *Salmonella typhimurium* and the significance of fatty acids. This study mainly examined whether the anti-proteolytic peptide R7I improved fatty acid absorption impairment in the presence of *Salmonella typhimurium* infection and its protective effect on the jejunal intestinal barrier. The anti-proteolytic peptide R7I is distinguished by the feature that it is difficult to acquire drug resistance, and this work offers a fresh idea to address the public safety issue of *Salmonella* drug resistance. Additionally, it serves as a reference for the clinical use of in vivo antimicrobial peptides.

## 2. Results

### 2.1. R7I Could Inhibit the Development of Inflammatory Factors in Jejunal Tissue

The intestinal barrier is a crucial component that prevents hazardous and harmful intestinal microbes from entering the animal organism in addition to facilitating the conversion and absorption of nutrients. We first identified the inhibitory effect of R7I on inflammation because *Salmonella typhimurium* infection causes an inflammatory response that damages the intestinal barrier structure, which in turn limits substance absorption. The gene expression levels of inflammatory factors, including *IL-6*, *TNF-α*, and *IL-1β*, were assessed alongside the gene expression of key components involved in the recognition and signaling of lipopolysaccharide (LPS), such as *TLR4*, *MYD88*, and *NF-κB P65*. Remarkably, the R7I treatment group exhibited a substantial downregulation in the mRNA expression of *TLR4*, *MYD88*, *NF-κB P65*, as well as the inflammatory markers *IL-6*, *TNF-α*, and *IL-1β*, when compared to the control group (*p* < 0.05) (Figure 1). We chose the four genes that had the greatest changes in mRNA expression and checked to see if their differences in protein expression were equally substantial. The protein expression of TLR4, TNF-α, and IL-1β was significantly diminished in the R7I treatment group compared with the control group, and the phosphorylation level of p65, a key subunit of NF-κB involved in inflammation, was also found to be inhibited (*p* < 0.05) (Figure 1).

### 2.2. R7I Mitigates Oxidative Damage to the Gut Brought on by Salmonella typhimurium

Large numbers of reactive oxygen species are produced during the intestinal inflammatory process, and the buildup of reactive oxygen species causes oxidative stress, which worsens inflammation, creating a vicious cycle. The tight junction proteins are severely inhibited by oxidative stress, and intestinal permeability rises as a result, damaging the intestinal barrier. As a result, we examined R7I’s ability to reduce oxidative stress. The R7I group exhibited a significant decrease in MDA levels and a substantial increase in CAT, GSH-PX, and SOD activities compared to the control group (*p* < 0.05) (Figure 2). The expression levels of genes associated with antioxidants were also examined, revealing that R7I treatment significantly upregulated the *Nrf2* gene while considerably downregulating the expression of *Keap-1* and *I-NOS* genes (*p* < 0.05) (Figure 2). These findings suggested that R7I administration can alleviate oxidative damage in the intestinal tract of mice induced by *Salmonella typhimurium* infection.

### 2.3. Effects of R7I on Salmonella typhimurium-Induced Intestinal Barrier Damage

We also investigated R7I’s protective effects on the intestinal barrier in light of the aforementioned research findings that it could decrease the inflammatory response and reduce oxidative stress. We first observed the protective effect of R7I on the intestine by measuring the histological changes in the intestine. In the control group, congestion in the mucosal layer (red arrow) and breakage and absence of the lamina propria in the intestinal lumen (black arrow) were observed in the tissue sections. However, the R7I group exhibited a normal tissue structure with only minimal bleeding in the mucosal layer (red arrow). The blank group had structurally normal tissues without any lesions (Figure 3).

Considering the crucial role of tight junction proteins in maintaining intestinal barrier functionality, we further investigated the expression of these genes and proteins. The results showed that the expression of the ZO-1 at both gene and protein levels in the R7I treatment group was significantly higher than that in the control group (*p* < 0.05) (Figure 4). Notably, while the expression of the Claudin-1 gene was altered after *Salmonella typhimurium* treatment in the control group, R7I administration normalized its expression (Figure 4). These findings indicated that R7I preserved the integrity of the intestinal barrier and effectively reduced the damage caused by *Salmonella typhimurium* to the intestinal structures. The upregulation of ZO-1 expression and normalization of claudin-1 expression further support the protective effects of R7I on the intestinal barrier.

### 2.4. Alleviating Effect of R7I on Impaired Fatty Acid Absorption Caused by Salmonella typhimurium

Finally, we examined R7I’s impact on the recovery of fatty acid uptake capability. In blood biochemical assays, we observed notable differences in serum levels of free fatty acids, triglycerides, and total cholesterol between the control and blank groups. The control group exhibited significantly lower levels of these parameters compared to the blank group (*p* < 0.05). Conversely, the R7I group displayed significantly higher concentrations of serum-free fatty acids and triglycerides compared to the control group (*p* < 0.05) (Figure 5).

Furthermore, we measured the concentrations of lipid metabolites in the serum. The control group showed significantly decreased levels of high-density lipoprotein (HDL) and low-density lipoprotein (LDL) compared to the blank group (*p* < 0.05) (Figure 5), indicating that *Salmonella typhimurium* infection disrupts fatty acid metabolism. Excitingly, R7I treatment significantly improved this disruption (*p* < 0.05). Based on these results, it can be inferred that R7I administration could reduce the harmful effects of *Salmonella typhimurium* infection on fatty acid absorption and lipid metabolism.

### 2.5. R7I Enhances the Expression of Fatty Acid Uptake-Related Genes and Proteins in Salmonella typhimurium-Treated Mice

The aforementioned findings showed that R7I restored the capacity for fatty acid uptake, so we sought to further investigate if R7I had an impact on the expression of proteins involved in fatty acid uptake. Finally, we examined the expression of genes and proteins related to fatty acid absorption in the intestine. At the gene level, the R7I treatment group significantly outperformed the control group in the expression of genes such as *FABP1*, *FABP2*, *FATP4*, and *CD36* (*p* < 0.05) (Figure 6). We subsequently looked at the protein expression of FATP4 and FABP2, which are the most important proteins involved in fatty acid absorption in the jejunal region. In terms of protein expression, FATP4 and FABP2 also showed a significant increase in the R7I treatment group compared to the control group (*p* < 0.05) (Figure 6). These findings suggested that R7I administration effectively counteracts the negative impact of *Salmonella typhimurium* infection on the expression of key genes and proteins involved in fatty acid absorption. This demonstrates the potential of R7I as a therapeutic intervention to ensure proper intestinal absorption of fatty acids.

## 3. Discussion

Anti-proteolytic peptide R7I has the potential to replace antibiotics with broad-spectrum bactericidal properties and is a very effective alternative. Due to R7I’s ability to protect the intestinal barrier, maintain normal intestinal morphology, and kill harmful bacteria in the intestine thanks to its anti-enzymatic properties and effective bactericidal ability, it is possible to alleviate impaired absorption of substances caused by bacterial infections, particularly fatty acids.

The initial line of defense for natural immunity is built by TLR4, a significant member of the TOLL-like receptor family, which specifically identifies lipopolysaccharide, a component of Gram-negative bacteria’s cell walls [32]. LPS can interact with TLR4/MD2 to create a complex that specifically activates the MyD88-dependent pathway, which in turn activates the transcription factor NF-κB [33]. NF-κB, as a key signaling molecule, regulates numerous genes for different processes of the immune and inflammatory responses [34]. Inflammatory substances like IL-1β, IL-6, and TNF-α are released in huge quantities as a result of NF-κB activation, harming the animal’s body [35]. In this work, the TLR4/NF-κB signaling pathway was thought to have been activated by *Salmonella typhimurium* based on the dramatically increased expression of the TLR4 gene and its downstream genes MYD88 and NF-κB in the control group. This theory was further supported by the increased expression of inflammatory mediators like IL-1β, IL-6, and TNF-α. The intestinal epithelial barrier becomes dysfunctional due to the increased production of inflammatory agents and the release of endotoxin (LPS) by *Salmonella typhimurium* in the digestive tract, which also compromises the tight junctions’ structural integrity [16,17]. Reduced growth and systemic disease are the results of a weakened intestinal barrier that causes malabsorption of nutrients and the spread of harmful microorganisms from the gut to other organ systems [17,22,36]. Excitingly, R7I decreased the expression of inflammatory markers such as IL-1β, IL-6, and TNF-α by lowering the expression of genes and proteins connected to the TLR4/NF-κB signaling pathway. This may be one of the reasons why R7I protects the intestinal barrier and alleviates impaired fatty acid absorption.

In addition to ensuring the transformation and absorption of nutrients, the intestinal barrier is a crucial structure that guards against the entry of toxic and harmful intestinal microorganisms into the animal organism. Therefore, compared to other tissues, intestines that frequently come into contact with bacteria and viruses are more susceptible to oxidative stress [37,38]. It has been discovered that oxidative stress greatly inhibits tight junction proteins and raises intestinal permeability, both of which lead to damage to the intestinal barrier [39,40]. In this work, R7I increased intestinal antioxidant capacity and alleviated oxidative stress. R7I decreased the amount of intestinal MDA, inhibited high levels of Keap-1 and I-NOS gene expression, and enhanced the amount of intestinal CAT, SOD, and GSH-PX, as well as Nrf2 gene expression. In addition to acting as an antioxidant in vivo, Nrf2 is a key regulator of intracellular redox, which suppresses the production of inflammatory factors [41]. According to one study, NF-κB activity rises as a result of Nrf2 depletion, which intensifies the inflammatory response [42]. Increased inflammatory response, however, will support oxidative stress and intestinal barrier breakdown. In light of this, we suggest that the anti-proteolytic peptide R7I can reduce intestinal damage by reducing oxidative stress and enhancing antioxidant capacity, hence preserving the normal absorption of substances in the intestine. R7I effectively alleviated the reduction of Tight junction protein expression and the damage caused by *Salmonella typhimurium* to the intestinal structure in this study.

Fatty acids are one of the three major nutrients and one of the essential nutrients. Fatty acids participate in the composition of biofilms, control cell signaling, and are engaged in many physiological processes in the animal body. The triglycerides that are created after fatty acids are digested and absorbed combine with proteins to form celiac particles, which are then carried throughout the body via lymph and blood or enter the bloodstream as free fatty acids. In this work, R7I dramatically boosted the levels of serum triglycerides, free fatty acids, and lipid metabolites in mice infected with *Salmonella typhimurium*, indicating that R7I restored intestinal fatty acid absorption. The process of absorbing fatty acids is intricate. The majority of fatty acids in the food combine to create triglycerides with glycerol, which are then broken down by pancreatic lipase in the intestine into monoglycerides, diglycerides, free fatty acids, and glycerol, which are then absorbed by the intestinal epithelium. Although it is now well acknowledged that specific carrier proteins are necessary for intestinal fatty acid absorption, there is debate concerning this [43,44,45]. The fatty acid transporter protein family, the fatty acid binding protein family, and CD36 are the proteins connected to fatty acid uptake that were discovered in the current study. The results of this study demonstrated that *Salmonella typhimurium* significantly decreased the expression of FABP1, CD36, FABP2, and FATP4 in the intestine of mice that were infected with the bacteria. This suggests that *Salmonella typhimurium* caused fatty acid malabsorption in mice by preventing the expression of genes and proteins involved in fatty acid absorption. However, the anti-proteolytic peptide R7I restores the ability of the intestine to absorb fatty acids by increasing the expression of FABP2, FABP1, FATP4, and CD36.

## 4. Materials and Methods

### 4.1. Ethics Approval and Consent to Participate

The protocols used in this experiment were approved by the Northeast Agricultural University Institutional Animal Care and Use Committee. All the Animal care and treatment complied with the standards described in the “Laboratory Animal Management Regulations” (revised 2016) of Heilongjiang Province, China.

### 4.2. Synthesis and Characterization of Peptides

GL Biochem Corporation (Shanghai, China) produced R7I (IRPI IRPI IRPI IRPI IRPI IRPI-NH2) (Appendix A) and matrix-assisted laser desorption/ionization time-of-flight mass spectrometry and reversed-phase high-performance liquid chromatography was used to identify it (MALDI-TOF MS; LinearScientific Inc., Renault, NV, USA; RP-HPLC; SHIMADZU Inc., Kyoto, Japan). Matrix-assisted laser desorption/ionization time-of-flight mass spectrometry (MALDITOF MS) and reversed-phase high-performance liquid chromatography (RP-HPLC) analyses (Appendix A) indicated that the measured molecular weights of the peptides were close to their theoretical molecular weights, and the purities of the peptides were more than 95%, which indicates that the peptides were successfully synthesized. HPLC with a column of SHIMADZU Inertsil ODS-SP 4.6 × 250 mm × 5 µm, 214 nm, 10 μL column using a nonlinear water/acetonitrile gradient containing 0.1% Trifluoroacetic at a flow rate of 1.0 mL/min.

### 4.3. Construction of an Enteritis Model and Experimental Design

KUNMING male mice aged six weeks were supplied by Liaoning Changsheng Biotechnology Co., (Shenyang, China). All mice (20.00 ± 2.00 g, 6 weeks old) were separated into 3 groups (*n* = 8) at random and given food and environment training for 3 days. Under carefully regulated environmental conditions (relative humidity 40–60%; temperature 25 ± 2 °C; lighting cycle 12 h/day), the mice were given access to clean water and a regular laboratory diet throughout the experiment: 18% crude protein, 4% crude fat, 5% crude fiber, 8% ash, and 10% water made up the typical laboratory diet.

*Salmonella typhimurium* ATCC 14028 was obtained from the Institute of Animal Nutrition, Northeast Agricultural University (Harbin, Heilongjiang, China). We chose the proper bacterial concentration based on the outcomes of the preliminary tests. *Salmonella typhimurium* strains stored frozen at −20 °C were transferred to LB medium (37 °C, 220 r) for overnight growth. To attain the logarithmic growth period, the bacterial broth was transferred at a ratio of 1:100 to the fresh LB medium for 5–6 h. The bacterial solution was centrifuged, the precipitate was washed with saline three times, and then the precipitate was resuspended in saline with the colony count set to 4.4 × 10^9^ CFU/mL. Three groups of mice were created at random: a control group, an R7I group, and a blank group. For the first two days of the experiment, the blank group received 200 mL of saline gavage, while the other groups received 200 mL of *Salmonella typhimurium* 14,028 twice a day at 12-h intervals. On days three to four, R7I (20 mg/kg) was gavaged twice daily at 12-h intervals to the R7I group, whereas saline was administered in equal volumes to the blank and control groups. On day five, isoflurane-induced general anesthesia was used to euthanize all animals via cervical dislocation.

### 4.4. Collection of Blood, Tissues, and Organs Samples

The blood drawn was centrifuged at 3000 rpm for 10 min at 4 °C. For later testing, the supernatant was collected and kept at −80 °C. For hematoxylin and eosin (H&E) staining, the jejunum was sliced, and the pre-cooled 4% paraformaldehyde was added. Separately kept at −80 °C in cryogenic vials were the contents of the liver, small intestine, colon, and cecum.

### 4.5. Blood Biochemistry Analysis

Serum biochemical indices of high-density lipoprotein cholesterol (HDL-C), low-density lipoprotein cholesterol (LDL-C), total cholesterol (CHOL), triglycerides (TG), glucose (GLU) were measured using the automatic biochemical analyzer (Roche, Cobus-Mira-Plus, Roche Diagnostic System Inc., Basel, Switzerland).

### 4.6. Determination of Free Fatty Acid Content

According to the manufacturer’s instructions, kits made by Nanjing Jiancheng Institute of Biological Engineering were used to quantify the free fatty acid content of mouse serum, liver, and cecum contents.

### 4.7. The Antioxidant Function of the Empty Intestine

Total antioxidant capacity (T-AOC), superoxide dismutase (SOD), glutathione peroxidase (GSH-Px), malondialdehyde (MDA), and catalase (CAT) enzyme activities in the empty intestine were determined using commercially available kits (Nanjing Jiancheng Bioengineering Institute, Nanjing, China) according to the manufacturer is instructions.

### 4.8. Histological Analysis

Hematoxylin-eosin staining was used to accomplish the histological investigation (H&E). The jejunum was dried, embedded in paraffin blocks, and fixed with 4% paraformaldehyde-PBS overnight. The materials were subsequently divided into 5 μm slices, which underwent deparaffinization, hydration, and H&E staining for histologic analysis.

### 4.9. RNA Extraction and Real-Time Quantitative PCR

Briefly, total RNA was extracted with Trizol, and then RNA was reversely transcribed into cDNA after genomic DNA was removed by a two-step method according to the instructions of the kit. SYBR Green method was used to analyze the mRNA expression of specific primers for different genes. The reagents and kits were purchased from Nanjing Vazyme Biotech Co., Ltd., Nanjing, China. Subsequently, real-time PCR was performed using CFX Connect (Bio-Rad, Hercules, CA, USA). Primer 5.0 software was used to design primers for specific genes (Table 1). The internal control gene was β-actin. The relative abundance of target genes was calculated using the 2^−ΔΔCt^ approach. The expression of target gene mRNA in the control group was taken as the baseline relative to the treatment group (i.e., fold-change).

### 4.10. Assessment of Protein Expression by Western Blot Analysis

Total protein from the jejunum tissues was separated using commercial protein extraction reagents (Beyotime, Shanghai, China). Reducing SDS-PAGE electrophoresis was conducted to separate 40 μg protein, which was then transferred to PVDF membranes (Millipore, Billerica, MA, USA). Afterward, membranes were blocked with 5% skimmed milk for 2 h in Tris-Tween saline buffer, followed by incubation using primary antibodies. The HRP-conjugated secondary antibodies were subsequently incubated for 2 h at room temperature. The Alpha Imager 2200 V5.5 software (Alpha Innotech Corporation, San Leandro, CA, USA) was employed to develop protein blots. Lastly, protein signals were quantified digitally and normalized to the relative expression of β-actin. ImageJ 1.54f software (National Institutes of Health, Bethesda, MD, USA) was used to determine the band density. Internal reference β-actin (Art No. AA128) and horseradish peroxidase-labeled goat anti-rabbit IgG (Art No. A0208) from Shanghai Beyotime, China. FABP2 Antibody (Art No. 67691) from Wuhan proteintech, China Other antibodies from Wuhan ABcional, China (TLR4 Antibody, Art No. A11226; P-P65 Antibody, Art No. AP0124; TNF-α Antibody, Art No. A0277; IL-1β Antibody, Art No. A12688, FATP4 Antibody, Art No. A21969; ZO-1 Antibody, Art No. A0659; Occludin Antibody, Art No. A2601; Claudin-1 Antibody, Art No. A21971).

### 4.11. Statistical Analysis

We refer to Li’s statistical method. Results are reported as means ± standard errors of the mean (SEM) and evaluated with one-way analysis of variance (ANOVA). The Tukey test was used to detect differences among treatments. The SPSS V25 (SPSS Inc., Chicago, IL, USA) software was used for all statistical analyses. A *p*-value of <0.05 was considered statistically significant. All data were visualized using Graphpad Prism 8.0 (Graphpad Inc., San Diego, CA, USA).

## 5. Conclusions

Anti-proteolytic peptide R7I reduced the production of inflammatory factors such as IL-6, TNF-α, and IL-1β by inhibiting the expression of TLR4 and NF-κB, and reduced inflammation. In addition, by increasing the level of SOD and GSH-PX and decreasing the level of MDA, the anti-proteolytic peptide R7I could reduce oxidative stress and enhance antioxidant capacity. Meanwhile, anti-proteolytic peptide R7I also promotes the expression of ZO-1 protein to protect the intestinal barrier with normal intestinal morphology. By promoting the expression of genes or proteins associated with fatty acid absorption in the jejunum, such as FATP4, FABP2, and CD36, R7I restores fatty acid absorption in the intestine. Overall, the anti-proteolytic peptide R7I attenuated inflammation associated with *Salmonella typhimurium* infection, protected the intestinal barrier, and restored fatty acid absorption.

## Figures and Tables

**Figure 1 ijms-24-16409-f001:**
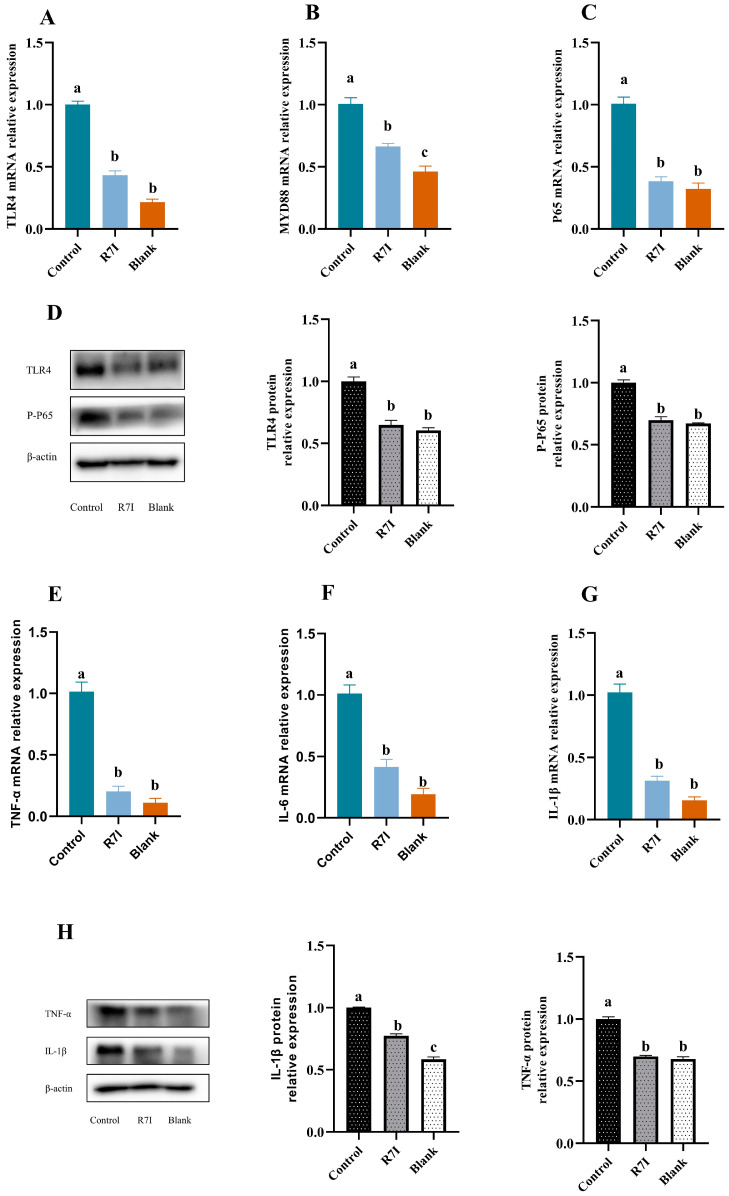
(**A**–**C**,**E**–**G**) Relative mRNA expression of TLR4, Myd88, NF-κB P65, and inflammatory factors in jejunal tissue of each group (*n* = 6). (**D**,**H**) Relative protein expression of jejunal TLR4, P-P65, TNF-α, L-1β, and other proteins was detected by Western blot (*n* = 3). A 20 μg sample of protein was taken. Beta-actin was employed as an internal control. β-Actin, beta-actin, 42 kDa; TLR4, Toll-like receptor 4; 96 kDa; P-P65, Phospho-NF-κB P65, 60 kDa; TNF-α, Tumor necrosis factor α, 26 kDa; IL-1β, Interleukin-1β, 31 kDa. The data were presented as mean ± SEM. One-way ANOVA with a Tukey post-test was used to determine statistical significance. In a bar chart, different lowercase letters indicate significance (*p* < 0.05).

**Figure 2 ijms-24-16409-f002:**
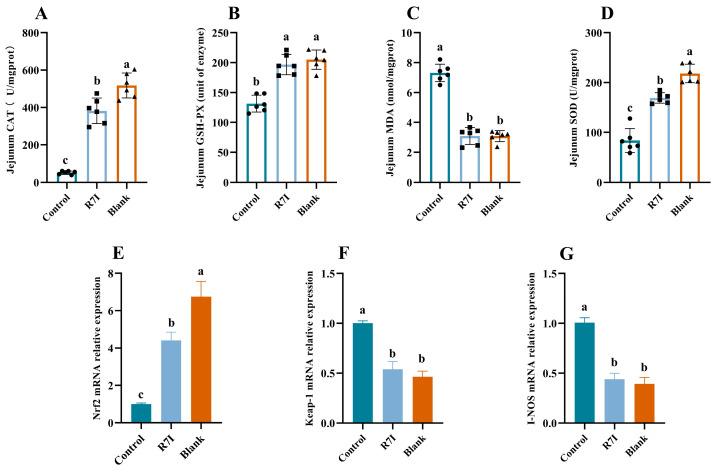
(**A**) Levels of catalase in the jejunum tissue of each group (*n* = 6). (**B**) Levels of glutathione peroxidase in the jejunum tissue of each group (*n* = 6). (**C**) Malondialdehyde level in the jejunum tissue of each group (*n* = 6). (**D**) Levels of superoxide dismutase in the jejunum tissue of each group (*n* = 6). (**E**–**G**) Relative mRNA expression of Nrf2, Keap-1, and I-NOS in the jejunum of each group, respectively (*n* = 6). The data were presented as mean ± SEM. One-way ANOVA with a Tukey post-test was used to determine statistical significance. In a bar chart, different lowercase letters indicate significance (*p* < 0.05).

**Figure 3 ijms-24-16409-f003:**
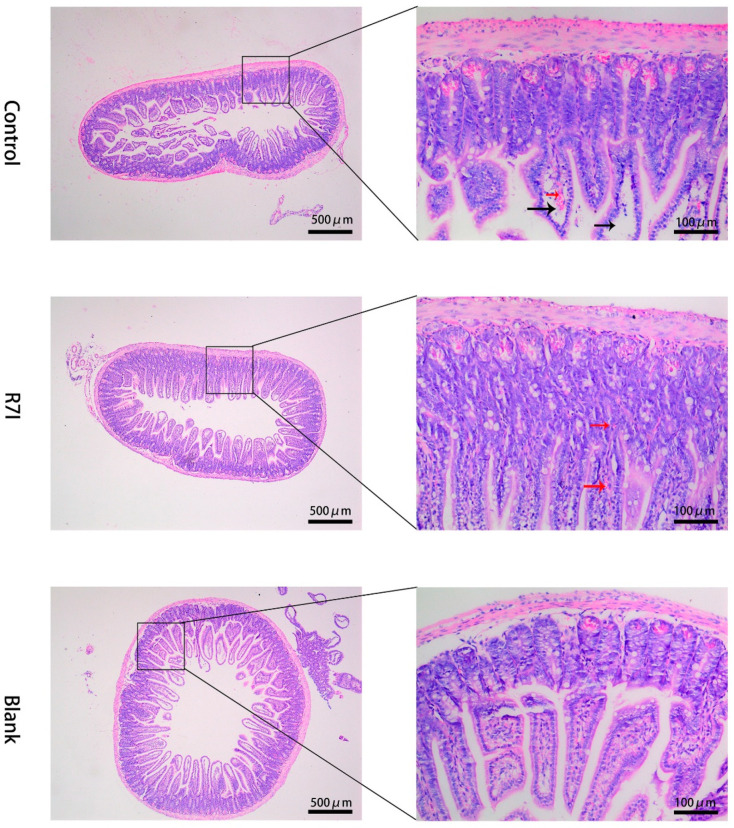
H&E-stained histological sections of intestinal tissues. Red arrows indicate congestion of the mucosal layer, and black arrows indicate disruption and absence of the lamina propria of the intestine.

**Figure 4 ijms-24-16409-f004:**
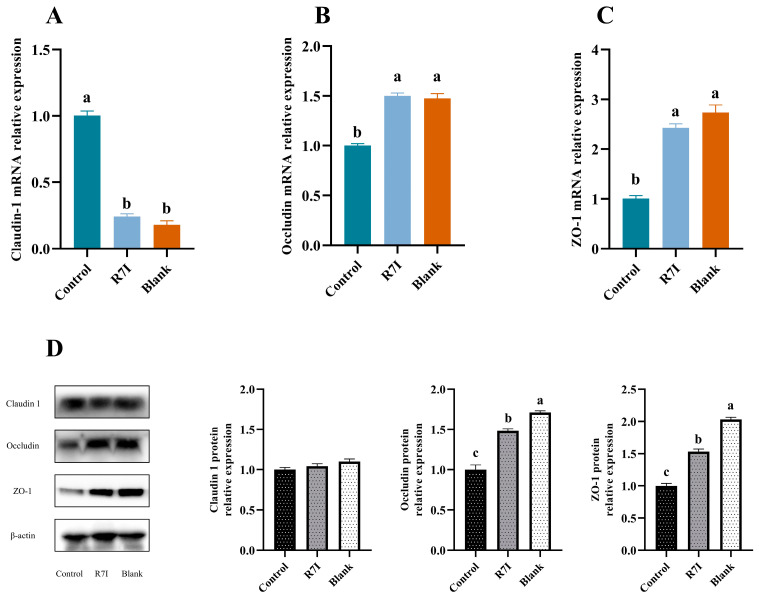
(**A**–**C**) Relative mRNA expression of tight junction proteins in each group of the jejunum (*n* = 6). (**D**) Relative protein expression of the tight junction proteins of the jejunum in the assay using Western blot (*n* = 3). A 20 μg sample of protein was taken. Beta-actin was employed as an internal control. β-Actin, beta-actin, 42 kDa; ZO-1, tight junction protein 1; 250 kDa; occludin, 65 kDa; claudin 1, 23 kDa. The data were presented as mean ± SEM. One-way ANOVA with a Tukey post-test was used to determine statistical significance. In a bar chart, different lowercase letters indicate significance (*p* < 0.05).

**Figure 5 ijms-24-16409-f005:**
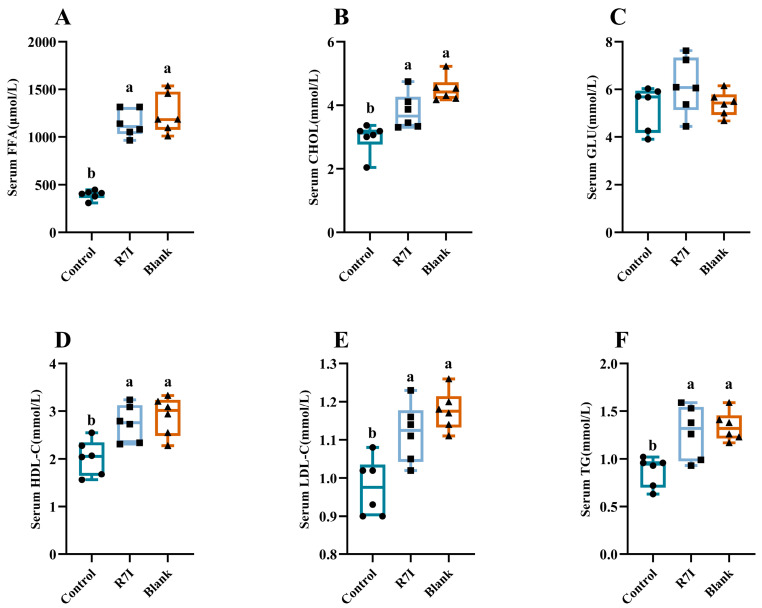
(**A**) Serum-free fatty acid content in each group (*n* = 6). (**B**) Serum total cholesterol content in each group (*n* = 6). (**C**) Serum glucose content in each group (*n* = 6). (**D**) Serum HDL cholesterol content in each group (*n* = 6). (**E**) Serum LDL cholesterol in each group (*n* = 6). (**F**) Serum triglyceride content in each group (*n* = 6). The data were presented as mean ± SEM. One-way ANOVA with a Tukey post-test was used to determine statistical significance. In a bar chart, different lowercase letters indicate significance (*p* < 0.05).

**Figure 6 ijms-24-16409-f006:**
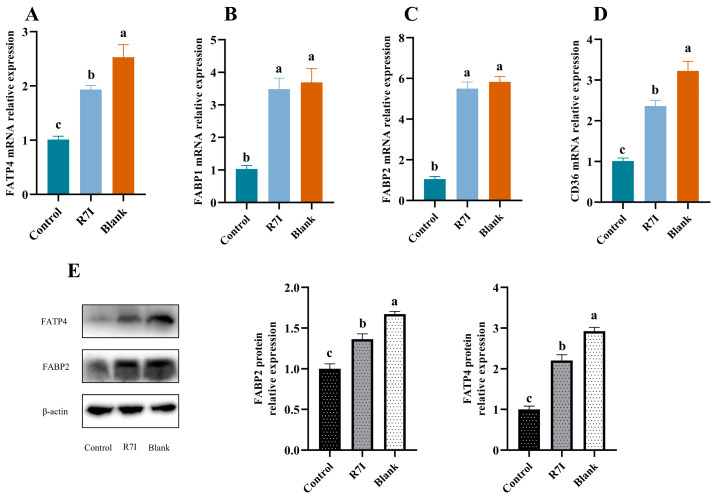
(**A**–**D**) Relative mRNA expression levels of fatty acid absorption-related genes in the jejunum of each group (*n* = 6). (**E**) Relative protein expression of FATP4 and FABP2 in the jejunum was detected using Western blot (*n* = 3). A 20 μg sample of protein was taken. Beta-actin was employed as an internal control. β-Actin, beta-actin, 42 kDa; FATP4, fatty acid transporter protein4, 65 kDa; FABP2, fatty acid binding protein 2; 15 kDa. The data were presented as mean ± SEM. One-way ANOVA with a Tukey post-test was used to determine statistical significance. In a bar chart, different lowercase letters indicate significance (*p* < 0.05).

**Table 1 ijms-24-16409-t001:** Primer sequences for q-PCR.

Gene	Sequences (5′→3′)	Fragments Sizes	Gen Bank No.
IL-6	F: CTCCCAACAGACCTGTCTATAC	97 bp	NM_031168.2
	R: CCATTGCACAACTCTTTTCTCA		
IL-1β	F: GAAATGCCACCTTTTGACAGTG	116 bp	NM_008361.4
	R: TGGATGCTCTCATCAGGACAG		
TNF-α	F: ATGTCTCAGCCTCTTCTCATTC	179 bp	NM_013693.3
	R: GCTTGTCACTCGAATTTTGAGA		
NF-κB-P65	F: AGACCCAGGAGTGTTCACAGACC	141 bp	NM_001402548.1
	R: GTCACCAGGCGAGTTATAGCTTCAG		
MYD88	F: TCATGTTCTCCATACCCTTGGT	175 bp	NM_010851.3
	R: AAACTGCGAGTGGGGTCAG		
TLR4	F: TCCCTGCATAGAGGTAGTTCC	119 bp	NM_021297.3
	R: TCAAGGGGTTGAAGCTCAGA		
NRF2	F: TAGATGACCATGAGTCGCTTGC	153 bp	NM_010902.5
	R: GCCAAACTTGCTCCATGTCC		
KEAP-1	F: TGCCCCTGTGGTCAAAGTG	104 bp	NM_001110307.1
	R: GGTTCGGTTACCGTCCTGC		
I-NOS	F: ACATCGACCCGTCCACAGTAT	177 bp	NM_001313922.1
	R: CAGAGGGGTAGGCTTGTCTC		
FATP4	F: ACCAGGGTGCCAACAACAAGAAG	121 bp	NM_011989.5
	R: GTGCGATCTCGGAAGTACAGGTAAC		
CD36	F: GCAGGTCTATCTACGCTGTGTTCG	111 bp	XM_030254088.1
	R: TGTCTGGATTCTGGAGGGGTGATG		
FABP1	F: AAGTGGTCCGCAATGAGTTCACC	87 bp	NM_017399.5
	R: CCAGCTTGACGACTGCCTTGAC		
FABP2	F: GATTGCTGTCCGAGAGGTTTCTGG	80 bp	NM_007980.3
	R: TAAAGAATCGCTTGGCCTCAACTCC		
Occludin	F: TGCTTCATCGCTTCCTTAGTAA	155 bp	NM_001360536.1
	R: GGGTTCACTCCCATTATGTACA		
ZO-1	F: CTGGTGAAGTCTCGGAAAAATG	97 bp	NM_009386.2
	R: CATCTCTTGCTGCCAAACTATC		
Claudin 1	F: AGATACAGTGCAAAGTCTTCGA	86 bp	NM_016674.4
	R: CAGGATGCCAATTACCATCAAG		
β-actin	F: CTACCTCATGAAGATCCTGACC	100 bp	NM_007393.5
	R: CACAGCTTCTCTTTGATGTCAC		

## Data Availability

All relevant data are either presented within the manuscript or made available online as Appendix A. The information is accessible upon reasonable request. The study’s raw data and associated analyses may be obtained from the corresponding author upon reasonable request.

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
