# Peer review of "Anti-Proteolytic Peptide R7I Protects the Intestinal Barrier and Alleviates Fatty Acid Malabsorption in Salmonella typhimurium-Infected Mice"

_ijms, 2023, doi:10.3390/ijms242216409_

Round 1

Reviewer 1 Report

Comments and Suggestions for Authors

Salmonella is a bacterium responsible for most food poisoning in humans, sometimes even resulting in death. Therefore, any research on the mechanism of their formation and the possibility of limiting their effects should be considered very important.

The introduction to the obtained results contains information relevant to the content of the article and was written based on new literature data. The research was designed in a thoughtful way, carried out using classic methods, and their results described logically and carefully.

I feel a certain dissatisfaction after reading subsection 4.2. It is written very concisely, and the information contained therein does not allow me to form an opinion about the structure of the peptide and its purity. In the description of the MALDI-TOF analysis, there is no information about the matrix used by the authors and the molecular ion obtained. In the case of RP HPLC analysis, there is no information about the separation parameters (mobile phase composition, flow or detector used). The authors report that the purity of the peptide used was >95%. How was it defined? Was this based on the response of the UV-Vis detector? There is also no information about the chromatograph used. Perhaps it would be good if the authors included a MALDI-TOF mass spectrum and an HPLC chromatogram in the supporting materials.

Author Response

Dear Reviewers:

Thank you for the comments concerning our manuscript entitled “Anti-proteolytic Peptide R7I protects the intestinal barrier and alleviates fatty acid malabsorption in Salmonella typhimurium-infected mice”.

Those comments are all valuable and very helpful for revising and improving our paper, as well as the important guiding significance to our research. We have studied comments carefully and have made corrections which we hope meet with approval. Revised portions are marked with different colors on the paper. The main corrections in the paper and the responses to the reviewer's comments are as follows:

  1. I feel a certain dissatisfaction after reading subsection 4.2. It is written very concisely, and the information contained therein does not allow me to form an opinion about the structure of the peptide and its purity. In the description of the MALDI-TOF analysis, there is no information about the matrix used by the authors and the molecular ion obtained. In the case of RP HPLC analysis, there is no information about the separation parameters (mobile phase composition, flow, or detector used). The authors report that the purity of the peptide used was >95%. How was it defined? Was this based on the response of the UV-Vis detector? There is also no information about the chromatograph used. Perhaps it would be good if the authors included a MALDI-TOF mass spectrum and an HPLC chromatogram in the supporting materials.

Response:Thank you for underlining this deficiency.The structure of R7I is shown in Figure S1. HPLC with a column of SHIMADZU Inertsil ODS-SP 4.6×250 mm×5 µm, 214 nm, 10 μL column using a nonlinear water/acetonitrile gradient containing 0.1% Trifluoroacetic at a flow rate of 1.0 mL/min. Matrix-assisted laser desorption/ionization time-of-flight mass spectrometry (MALDITOF MS) and reversed-phase high-performance liquid chromatography (RP-HPLC) analyses (Figures S2 and S3) indicated that the measured molecular weights of the peptides were close to their theoretical molecular weights, and the purities of the peptides were more than 95%, which indicates that the peptides were successfully synthesized. We have made changes and additions to lines 318 to 329 of the manuscript, which are redlined. “GL Biochem Corporation (Shanghai, China) produced R7I (IRPI IRPI IRPI IRPI IRPI IRPI-NH2) (Figures S1), and matrix-assisted laser desorption/ionization time-of-flight mass spectrometry and reversed-phase high-performance liquid chro-matography was used to identify it (MALDI-TOF MS; LinearScientific Inc, RP-HPLC; SHIMADZU Inc). Matrix-assisted laser desorption/ionization time-of-flight mass spectrometry (MALDITOF MS) and reversed-phase high-performance liquid chroma-tography (RP-HPLC) analyses (Figures S2 and S3) indicated that the measured molec-ular weights of the peptides were close to their theoretical molecular weights, and the purities of the peptides were more than 95%, which indicates that the peptides were successfully synthesized. HPLC with a column of SHIMADZU Inertsil ODS-SP 4.6×250 mm×5 µm, 214 nm, 10 μL column using a nonlinear water/acetonitrile gradient con-taining 0.1% Trifluoroacetic at a flow rate of 1.0 mL/min.” Also, we will add Figures S1, S2, and S3 to the supplementary material.

Reviewer 2 Report

Comments and Suggestions for Authors

The research is investigating the protective effects of Anti-proteolytic Peptide R7I on the gut during Salmonella typhimurium infection. It appears that R7I has several positive effects on the intestine, including reducing inflammation, preserving the antioxidant balance, maintaining intestinal structure, and restoring the expression of specific proteins involved in fatty acid absorption.

While the findings of the article suggest that R7I has potential benefits in mitigating the damage caused by Salmonella typhimurium infection, there are a few points that could be addressed or elaborated on for a more comprehensive understanding of the research

How does Anti-proteolytic Peptide R7I exert these protective effects? Understanding the molecular mechanisms behind its actions could provide valuable insights into its potential as an alternative to antibiotics.

More details about the experimental methods used in the study, such as how R7I was administered, the duration of the experiment, and the dosage, would be essential for evaluating the validity and reproducibility of the results.

It's important to understand the characteristics of the control group used in the study. What was the basis for comparison, and were there any unexpected observations in the control group that might affect the interpretation of the results?

Information about the sample size and the statistical methods employed is crucial. A larger sample size generally provides more reliable results. Additionally, the specific statistical tests used and the significance threshold (P-value) are essential for assessing the strength of the findings.

Did the study assess the long-term effects of R7I treatment? Understanding whether the positive effects observed are sustained over time would be valuable in evaluating its potential as a long-term solution.

Based on the findings, what are the potential future research directions? Are there specific aspects that need further investigation or areas where R7I could be applied clinically?

Comments on the Quality of English Language

Quality of English is questionable, there are a few syntax and grammar errors in the text.

Author Response

Dear Reviewers:

Thank you for the comments concerning our manuscript entitled “Anti-proteolytic Peptide R7I protects the intestinal barrier and alleviates fatty acid malabsorption in Salmonella typhimurium-infected mice”.

Those comments are all valuable and very helpful for revising and improving our paper, as well as the important guiding significance to our research. We have studied comments carefully and have made corrections which we hope meet with approval. Revised portions are marked with different colors on the paper. The main corrections in the paper and the responses to the reviewer's comments are as follows:

  1. How does Anti-proteolytic Peptide R7I exert these protective effects? Understanding the molecular mechanisms behind its actions could provide valuable insights into its potential as an alternative to antibiotics.

Response:Thank you for underlining this deficiency. In terms of inflammation R7I inhibited the TLR4/NF-signaling pathway by suppressing the expression of genes and proteins such as TLR4, MYD88, and P65, which in turn inhibited the expression of inflammatory factors.R7I reduced intestinal MDA content, suppressed the high level of expression of the Keap-1 and I-NOS genes, and enhanced the intestinal CAT, SOD, and GSH-PX, as well as the Nrf2 gene expression. Through the above pathways, R7I increased intestinal antioxidant capacity and attenuated oxidative stress. we suggest that the Anti- proteolytic Peptide R7I can reduce intestinal damage by reducing oxidative stress and enhancing antioxidant capacity, hence preserving the normal absorption of substances in the intestine. R7I effectively alleviated the reduction of Tight junction protein expression and the damage caused by Salmonella typhimurium to the intestinal structure in this study. R7I increased the expression of fatty acid absorption-related genes such as FABP2, FABP1, FATP4 and CD36. which in turn restored the ability of the intestine to absorb fatty acids. These data provide strong guarantee for antimicrobial peptides to replace antibiotics

  1. More details about the experimental methods used in the study, such as how R7I was administered, the duration of the experiment, and the dosage, would be essential for evaluating the validity and reproducibility of the results.

Response:Thank you for underlining this deficiency. R7I was administered by gavage, and R7I was administered over two days, twice daily at 12-hour intervals, at a dose of 20 mg/kg. There were a total of three randomly created groups of mice in this experiment: the control group, the R7I group, and the blank group. For the first two days of the experiment, the blank group received 200 mL of saline gavage, while the other groups received 200 mL of Salmonella typhimurium 14028 twice a day at 12-hour intervals. On days three to four, R7I (20 mg/kg) was gavaged twice daily at 12-hour intervals to the R7I group, whereas saline was administered in equal volumes to the blank and control groups. On day five, isoflurane-induced general anesthesia was used to euthanize all animals via cervical dislocation. We have made changes and additions to lines 346 to 352 of the manuscript, which are redlined. “For the first two days of the experiment, the blank group received 200 mL of saline ga-vage, while the other groups received 200 mL of Salmonella typhimurium 14028 twice a day at 12-hour intervals. On days three to four, R7I (20 mg/kg) was gavaged twice daily at 12-hour intervals to the R7I group, whereas saline was administered in equal volumes to the blank and control groups. On day five, isoflurane-induced general an-esthesia was used to euthanize all animals via cervical dislocation.”

  1. It's important to understand the characteristics of the control group used in the study. What was the basis for comparison, and were there any unexpected observations in the control group that might affect the interpretation of the results?

Response:Thank you for underlining this deficiency. The setting of the control group was very scientific. The control group, like the treatment group, consisted of randomly grouped 6-week-old Kunming male mice, and both groups were exposed to the same environment, and both groups were gavaged with Salmonella typhimurium for the first two days of the experiment, with the only difference being that the R7I group was treated with R7I on the 3rd-4th day, and the control group was gavaged with the same volume of saline. This method excludes the influence of external factors.

  1. Information about the sample size and the statistical methods employed is crucial. A larger sample size generally provides more reliable results. Additionally, the specific statistical tests used and the significance threshold (P-value) are essential for assessing the strength of the findings.

Response:Thank you for underlining this deficiency. Your suggestion is very professional, due to the limited conditions in this experiment the sample size is not sufficient, in the subsequent experiments we will adopt your suggestions to improve the sample size of the experiment. In addition, we used spss software to analyze the data in this experiment, and the significance threshold was 0.05.

  1. Did the study assess the long-term effects of R7I treatment? Understanding whether the positive effects observed are sustained over time would be valuable in evaluating its potential as a long-term solution.

Response:Thank you for underlining this deficiency. This study is still in the initial phase of in vivo studies of R7I, and we have just concluded in vitro studies of R7I. We are positive about the long-term therapeutic efficacy of R7I, and we will evaluate its long-term therapeutic efficacy in future studies to demonstrate the potential of R7I as an alternative to antibiotics.

  1. Based on the findings, what are the potential future research directions? Are there specific aspects that need further investigation or areas where R7I could be applied clinically?

Response:Thank you for underlining this deficiency. In the future, we will further investigate the long-term therapeutic effects of R7I and assess whether it causes elevated bacterial resistance in vivo. In terms of substance absorption, we will further investigate whether R7I has a significant effect on the absorption of peptides and amino acids in the intestinal tract of animals. We also plan to use large animals, such as pigs, as experimental animals to investigate the therapeutic effects of R7I.

Grammatical problems in the text have been corrected.

Special thanks to you for your comments and suggestions. We did our best to improve the manuscript and these changes do not affect the content or framework of the paper. We sincerely thank the reviewers for their hard work and hope that the corrections will be approved. Thank you again for your comments and suggestions.
